# Clinical-Radiological Mismatch in Multiple Sclerosis Patients during Acute Relapse: Discrepancy between Clinical Symptoms and Active, Topographically Fitting MRI Lesions

**DOI:** 10.3390/jcm12030739

**Published:** 2023-01-17

**Authors:** Jutta Dünschede, Christoph Ruschil, Benjamin Bender, Annerose Mengel, Tobias Lindig, Ulf Ziemann, Markus C. Kowarik

**Affiliations:** 1Department of Neurology & Stroke, Hertie-Institute for Clinical Brain Research, Eberhard-Karls University of Tübingen, 72076 Tübingen, Germany; 2Department of Neuroradiology, Eberhard-Karls University of Tübingen, 72076 Tübingen, Germany

**Keywords:** multiple sclerosis, disease activity, relapse, magnetic resonance imaging (MRI), clinical-radiological mismatch

## Abstract

Background: Relapses in multiple sclerosis (MS) patients are usually defined as subacute clinical symptoms that last for at least 24 h. To validate a clinical relapse on magnetic resonance imaging (MRI), an anatomically fitting lesion with gadolinium enhancement in the central nervous system (CNS) would be mandatory. The aim of this study was to validate clinical relapses in regard to the concomitant detection of active, anatomically fitting MRI lesions. Methods: We performed a retrospective analysis of 199 MS patients with acute relapse who had received an MRI scan before the initiation of methylprednisolone (MPS) therapy. Clinical data and MRIs were systematically reanalyzed by correlating clinical symptoms with their anatomical representation in the CNS. Patients were then categorized into subgroups with a clinical-radiological match (group 1) or clinical-radiological mismatch (group 2) between symptoms and active, topographically fitting lesions and further analyzed in regard to clinical characteristics. Results: In 43% of our patients, we observed a clinical-radiological mismatch (group 2). Further analysis of patient characteristics showed that these patients were significantly older at the time of relapse. MS patients in group 2 also showed a significantly longer disease duration and significantly more previous relapses when compared to group 1. Comparing symptom clusters, the appearance of motor dysfunction during the current relapse was significantly more frequent in group 2 than in group 1. The overall dose of MPS treatment was significantly lower in group 2 than in group 1 with a similar treatment response in both groups. Conclusions: The substantial clinical-radiological mismatch during acute relapse in our study could be explained by several factors, including a psychosomatic component or disturbance of network connectivity. Alternatively, secondary progression or a diffuse neuro-inflammatory process might cause clinical symptoms, especially in older patients with a longer disease duration. As a consequence, treatment of clinical relapses and the definition of breakthrough disease should be reconsidered in regard to combined clinical and MRI criteria and/or additional biomarkers. Further studies are necessary to address the contribution of diffuse neuro-inflammation to the clinical presentation of symptoms.

## 1. Introduction

Relapses in multiple sclerosis (MS) patients are usually defined as subacute new or worsening clinical symptoms that last for at least 24 h and are separated from a previous attack by a minimum of 30 days. In addition, symptoms should not be attributable to confounding clinical factors such as fever, infection, injury or adverse reactions to concomitant medications [1,2]. In order to validate a clinical relapse on magnetic resonance imaging (MRI), an anatomically fitting lesion with gadolinium enhancement in the brain, spinal cord or both would be expected from a mechanistic point of view. Although relapses can often be linked to MRI lesions, there is a considerable mismatch between clinical symptoms and the occurrence of MRI lesions during the daily routine.

The validation of clinical relapses not only has implications for the decision of whether to treat a clinical relapse, escalate immunomodulatory treatments and define the disease course of MS, but also has to be reconsidered in the context of clinical studies. The first key endpoint in the majority of phase III clinical trials [3,4,5,6,7] in relapsing MS has been the annualized relapse rate and “relapse” has been defined by the patients’ clinical symptoms and their evaluation by a neurologist. To the best of our knowledge, a correlation analysis in regard to clinical symptoms and an active, anatomically fitting MRI lesion has not been performed in those studies. A new evolving concept to monitor disease activity in MS comprises the approach of “no evidence of disease activity” (NEDA) linking the absence of relapses with other measures such as MRI activity [2]. Although this approach helps to define stable disease phases, current definitions do not clearly combine clinical and MRI criteria in order to validate disease activity and/or relapses. In order to further evaluate clinical relapses in regard to the concomitant detection of MRI lesions, we correlated clinical symptoms with the neuro-anatomical location of active MRI lesions in a cohort of 199 MS patients with acute clinical relapse who had received an MRI scan before initiation of methylprednisolone (MPS) treatment.

## 2. Materials and Methods

### 2.1. Study Design and Patient Population

This study was approved by the ethics committee of the medical faculty of the Universität Tübingen. We performed a retrospective analysis of patients with MS who visited our clinic between 2018 to 2020; standardized MRI protocols had been implemented since the beginning of 2018. Study inclusion criteria for the MS patients were as follows: (1) diagnosis of clinically definite MS according to the 2017 [1] and 2010 [8] McDonald criteria, (2) acute relapse with clinical symptoms >24 h and (3) an MRI scan performed before the application of MPS. Exclusion criteria included (1) other CNS disease in addition to MS, (2) primary progressive form of MS, (3) additional relapse within 30 days prior to the actual relapse, (4) paroxysmal or fleeting symptoms, (5) infection and/or fever within 30 days prior to the actual relapse and (6) adverse reactions to concomitant medications. The patients’ symptoms and expanded disability status scale (EDSS) were obtained by a physician assistant and confirmed by a senior physician. In total, 217 patients with relapsing remitting MS (RRMS) and an acute clinical relapse were screened for this retrospective analysis. In 14 patients, no MRI data was available, and 4 patients received an MRI scan after starting MPS treatment; these patients were excluded from the cohort.

### 2.2. Correlation of Clinical Symptoms and MRI Lesions

Clinical symptoms of MS patients during relapse were reevaluated according to the standardized clinical reports and classified into cranial nerves symptoms, brain stem symptoms, motoric and sensory symptoms, coordination disturbances, bladder/bowel dysfunction and other symptoms.

MRIs were taken within 17 days in average after the first symptoms occurred and acquired with 3 T and 1.5 T scanners according to a standardized protocol that consists of a native 3D T1 MPRAGE (1 mm isotropic, 3 T: TR/TI 2300/900 ms, FA 8°, 1.5 T: TR/TI 1280/660 ms FA 15°), 3D Double Inversion Recovery (DIR, TR/TI1/TI2 7500/3000/450 ms, 1.5 T: 1.33 mm isotropic, TE 337 ms, 3 T: 1 mm isotropic, TE 392 ms), post contrast 3D T2-FLAIR (1 mm isotropic, 3 T: TR/TI/TE 7000/2050/392 ms, 1.5 T: TR/TI/TE 5000/1800/337 ms) and T1 MPRAGE (same protocol as precontrast), as well as additional diffusion-weighted images and 3 mm axial T2-TSE sequences that cover the posterior fossa and orbit (in-plane resolution 3 T: 0.6 × 0.5 mm^2^, 1.5 T: 0.9 × 0.7 mm^2^). A contrast agent was administered in a standardized fashion, with 0.1 mL/kg body weight 1 mmol/mL Gadobutrol (Gadovist^®^, Bayer Vital, Leverkusen, Germany) at a flow rate of 0.5 mL/s and a 20 mL saline flush after the 3D DIR image. The T1 MPRAGE with contrast enhancement is acquired approximately 10 min after administration of the contrast agent. The majority of patients in our cohort had received a cranial MRI and spinal MRI at the same time (81.4%). A total of 56.8% of the patients underwent MRI acquired with 3 T, 43.2% with 1.5 T. T2, T2-FLAIR and T1 with gadolinium enhancement images were analyzed using a standardized protocol by an experienced neuroradiologist during routine clinical reporting. These MRI reports were reanalyzed by correlating clinical symptoms and their anatomical representation in the human brain [9,10] with new gadolinium-enhancing (active) lesions by an experienced neurologist. Whenever this description was unclear or a discrepancy between clinical symptoms and MRI lesions was found, original MRI images were reanalyzed by two experienced neuroradiologists to validate the results.

According to this analysis, patients were categorized into two cohorts determining patients with (group 1) or without (group 2) a match between clinical symptoms and active MRI lesions. The time between first symptoms and MRI acquisition was 18 days in group 1 and 17 days in group 2 (no significant difference). Group 2, showing a clinical-radiological mismatch, was further divided into patients with gadolinium-enhancing but non-topographically fitting lesions (group 2a), patients with potentially fitting lesions without gadolinium enhancement (groups 2b) and patients without anatomically fitting lesions nor gadolinium enhancement lesions at all (group 2c). Data from patient group 2 was reevaluated in regard to potentially missing spinal MRI scans, which could have explained symptoms by active spinal cord lesions. Although this possibility was rather unlikely based on the clinical evaluation of each individual patient, spinal MRI scans were partially missing in group 2a (9 patients) and group 2b (5 patients); none were missing in group 2c. Although we implemented a standardized MRI protocol for the MS patients in our clinic, 24 of our patients (12.1%) did not receive these standardized MRI sequences, mostly due to unknown initial diagnosis. In detail, 14.2% of these patients were found in group 1, 2 patients (2.3%) in group 2a, 5 patients (5.8%) in group 2b and 1 patient (1.2%) in group.

Patient groups were further analyzed according to clinical characteristics such as disease duration, age at relapse, MPS treatment, treatment response, EDSS, immunomodulatory treatments and previous medication.

### 2.3. Data Analysis

Data analysis was performed in IBM SPSS statistics (version 27) and Graph Pad Prism (version 9.4.1). Because some of the parameters did not show a normal distribution, we used non-parametric tests for statistical analyses. Wilcoxon signed-rank tests and Kruskal–Wallis tests were used for the comparison of characteristics between patients with and without a clinical-radiological mismatch. Correlation matrix (Spearman) and principal component analysis were performed to further determine the most important factors that would discriminate between group 1 and group 2 (dependent variable) including the following clinical factors: gender, age at relapse, age at diagnosis, first diagnosis (during current relapse), disease duration, previous relapse, previous treatment, motor symptoms (only significant symptom), EDSS, previous treatment and MPS dosage. Two principal components (PCs) were extracted based on parallel analysis.

## 3. Results

### 3.1. Overall Patient Characteristics

Our MS patient collective showed a typical female to male ratio of 73% female and 27% male patients. The mean age at disease onset was 28 years and the mean disease duration was 5.9 years. During the acute relapse, 58% of the patients showed an EDSS score < 3 points, and the affected functional systems comprised sensory dysfunction in 54.8% of the patients, followed by brain stem/cranial nerve symptoms in 39.2% (including optic neuritis) and motoric dysfunction in 31.7% of the patients. In 41% of the patients, MS diagnosis was established with the current relapse so that only 52.5% of the patients received disease modifying therapies at the time-point of analysis. Further details on the patient collective are displayed in Table 1.

### 3.2. Clinical-Radiological Mismatch in Multiple Sclerosis Patients during Acute Relapse

Overall, 57% of the patients (113 MS patients; group 1) showed a match between clinical symptoms and anatomically fitting, gadolinium-enhancing lesions on MRI, whereas 43% of our patients (86 MS patients; group 2) showed no detectable topographic matches between the patients’ symptoms and active MRI lesions.

Regarding patient characteristics in group 1 and group 2 (Table 1, Figure 1), MS patients in group 2 were significantly older (35.2 years) at the time of relapse when compared to group 1 (31.2 years). These patients (group 2) showed a significantly longer disease duration with nearly 105 months compared to 45 months in group 1 and significantly more previous relapses when compared to group 1. Over 75% of the patients in group 2 suffered of more than 2 relapses before the current event, compared to only 43% in group 1 (Table 1, Figure 1). Comparing symptom clusters, the appearance of motor dysfunction during relapse was significantly higher in group 2 than in group 1, while other functional systems did not show significant differences. In more than 50% of the patients in both groups, only one functional system had been affected and deficits in the sensory system occurred most frequently. No significant differences could be shown in the EDSS score during the current relapse. Nearly 60% of the patients in both groups presented with an EDSS better than 3.0.

In order to minimize effects from immunosenescence [11] in our patient cohort, we performed a sub-analysis excluding patients >55 years (*n* = 10) and observed similar results.

We also compared patients who received an MRI acquired with 3.0 T to those who received a 1.5 T MRI. As the number of patients with 3.0 T MRI was nearly identical in both groups (43.4% in group 1 vs. 43.0% in group 2), the analysis did not show any significant differences.

### 3.3. Treatment of Acute Relapse and Disease Modifying Therapies

A total of 95% of the patients were treated with high-dose, short-term glucocorticoids, i.e., MPS (group 1: 98.2%; group 2: 90.7%). Ten patients did not receive drug therapy at all (group 1: 2 (1.8%); group 2: 8 (9.3%)), which indicated a significant difference (*p* = 0.016). Nine patients underwent plasma exchange (group 1: 7 (6.2%); group 2: 2 (2.3%)). The overall dose of intravenous MPS treatment during the current relapse was significantly shorter in group 2 then in group 1 (Table 1). Significantly more patients in group 2 (47.7%) received a low total dose of MPS (3000 mg) compared to patients in group 1 (24.8%). Vice versa, significantly more patients in group 1 received 5000 mg MPS in total (52.2%) compared to group 2 (24.4%). A marked improvement of clinical symptoms following MPS treatment, defined as patient reported outcome and/or improved clinical examination according to the final reports, was evident in both groups with 90% in group 1 and 84% in group 2.

Concerning previous treatments with disease modifying therapies (DMT), we only compared MS patients whose diagnosis had already been established before the current relapse. Patients in group 2 were slightly more often under DMT (58.2%) than patients in group 1 (45.1%), but this difference was not significant. In both groups, the current DMT had been taken on average for 27 months and patients in group 2 were treated more often with highly effective medications, i.e., ocrelizumab, cladribine, natalizumab, and fingolimod (53% vs. 44% in group 1); again, differences did not reach significance.

### 3.4. Subgroup Analysis in Patients with a Clinical-Radiological Mismatch (Group 2)

MS patients from group 2 (86 patients) who showed a clinical-radiological mismatch were further divided into 21 patients (group 2a) with new, gadolinium (Gd)-enhancing lesions that did not topographically match the symptoms, 34 patients that did display potentially fitting lesions that were neither new nor Gd-enhancing (group 2b), and 31 patients who did not show new lesions and the existing ones did not explain the symptoms (group 2c).

When comparing patients with non-fitting Gd-enhancing lesions (group 2a) with patients who showed potentially fitting lesions without Gd enhancement (group 2b), no significant differences were observed with regard to patient characteristics. However, patients without Gd enhancement and non-fitting lesions (group 2c) showed significantly more relapses (*p* = 0.003), a significantly lower percentage of patients with initial MS diagnosis during current relapse (*p* = 0.035) and a significantly lower MPS dosage (*p* = 0.012) compared to patients with non-fitting Gd-enhancing lesions (group 2a). Again, no significant differences were observed regarding age, disease duration, functional systems or EDSS. When comparing patients with potentially fitting lesions without Gd enhancement (group 2b) to patients without Gd enhancement/non-fitting lesions (group 2c), no significant differences were observed. However, the sample sizes in these subgroups were rather small, and potential differences might not have reached significance.

### 3.5. Multiple Variant Analyses

In order to further weight and group the influence of the clinical factors to the differentiation into group 1 and group 2, we additionally performed correlation and principal component analyses (Figure 2, Appendix A). A significant correlation with group 1/group 2 (match–mismatch) and age at relapse, first diagnosis (at current relapse), disease duration, previous relapse, motoric symptoms, previous treatment and MPS dosage was detectable; however, r values were all below 0.4. Multiple other correlations were found between the other clinical factors, such as disease duration and previous relapse, and further details are displayed in Figure 2a,b. Two principal components (PC1 and PC2) were extracted in our PC model, which explain 48.3% of the variance (PC1—29.6%, PCA2—18.7%). Major contributors for PC1 were disease duration, first diagnosis (current relapse lead the diagnosis of MS) and previous relapse; for PC2, the major factors were age at diagnosis, age at relapse and previous treatment.

## 4. Discussion

The definition of relapse in MS is highly relevant regarding the acute treatment, the application and escalation of DMTs, as well as for assessing the efficacy of drugs in clinical trials [12,13]. With MRI as the most reliable and sufficient tool to validate MS disease activity [14], we were interested in studying to what extent we could match clinical symptoms with active, neuroanatomically fitting lesions on MRI scans in MS patients with clinical relapse.

Overall, we found a considerably high number with 86 out of 199 MS patients in total (43%) showing a mismatch between clinical symptoms and active, topographically fitting MRI lesions (group 2). Regarding further patient characteristics, these patients were older at the time of relapse, had a longer disease duration with more previous relapses and the overall dose of MPS treatment was lower when compared to patients in whom symptoms and MRI lesions could be matched (group 1). With regard to the affected functional systems during relapse, the appearance of motor dysfunction was higher in group 2. To the best of our knowledge, association studies between clinical MS symptoms and MRI lesion mapping have mainly been performed in patients with intranuclear ophthalmoplegia so far [15,16,17]. Similar to our results, a mismatch between the symptoms of an intranuclear ophthalmoplegia and a fitting lesion in the medial longitudinal fasciculus was found in around 25% of the patients [15,16]; again, these cases were characterized by an older age, a longer disease duration, a higher EDSS and, more frequently, a progressive disease course [16]. In contrast to our study, the temporal correlation between acute symptoms and corresponding active MRI lesions has not been investigated in those studies because they were not performed during acute relapse.

In general, the clinical-radiological mismatch in our study could be explained by several factors including paroxysmal symptoms, pseudo-relapses, disturbance of network connectivity, the clinical manifestation of secondary progression or diffuse neuro-inflammatory processes. As an alternative explanation, alterations in self-perception or a psychosomatic component could also contribute to the observed mismatch, but seem difficult to be further differentiated in a retrospective approach. In MS, paroxysmal symptoms are mostly described as a sudden recurrence or intensification of symptoms such as spasms or pain episodes [12,18], whereas pseudo-relapses are usually defined as a return of symptoms during pro-inflammatory states unrelated to the autoimmune disorder [13]. Because several factors including body temperature can influence the ability of an action potential to propagate along an axon [19], fever, heat exposures or exercise can lead to the worsening of previous symptoms during pseudo-relapses. Due to the strict definition of relapse (new symptom for at least 24 h and 30 days after previous relapse) and the reevaluation of patients’ records, we can rule out paroxysmal symptoms and pseudo-relapses with a high probability in our cohort.

Another explanation for the mismatch between clinical symptoms and active MRI lesions would be disturbances in the functional connectivity through new active lesions that are not primarily associated with the topographically fitting functional brain region. It could be shown that functional connectivity between deep grey matter regions and the rest of the brain differs between MS patients and healthy controls, and changes in cognitive and motor performance are often associated with alterations in network connectivity [20]. Thus, an active lesion that does not primarily show a topographic match with the patient’s symptoms could also account for a new symptom by disturbing the functional connectivity, especially in a pre-injured brain. In our cohort, 21 patients showed Gd-enhancing but non-topographically fitting lesions that could contribute to disturbances in functional connectivity; however, to what extent these lesions indeed contribute to each patients’ individual symptoms remains difficult to determine.

Lastly, diffuse neuro-inflammation (including smoldering lesions, meningeal infiltrates) and cortical pathology [21] might contribute to the mismatch, especially in older MS patients with a longer disease duration. We observed a bimodal age distribution in patients with a clinical and MRI lesion mismatch (group 2), with a second age peak >45 years, which points towards a possibly age-related mechanism that might influence clinical symptoms. The results from our principal component analysis further support this observation because age at relapse, age at diagnosis and first diagnosis at current relapse were major factors in the differentiation between group 1 and 2. Diffuse neuro-inflammatory mechanisms include smoldering lesions, which can be visualized by paramagnetic iron rims on MRI [22]. Although disability scores tend to be worse in patients with paramagnetic iron rims, the exact clinical significance of rim lesions in MS is still unclear [23]. Furthermore, cortical lesions and meningeal infiltrates have been associated with progressive MS although gray matter damage can also begin during an earlier disease phase [24]. It could be shown that subpial cortical lesions are often associated with nearby meningeal lymphoid follicles, which are considered to represent ectopic CNS lymphoid structures that attract and maintain B and T lymphocytes [25,26]. A possible association between meningeal inflammation/cortical lesion and relapse is further supported in the rodent experimental autoimmune encephalomyelitis model, in which B cell accumulation in the subarachnoid space and meningeal inflammation is associated with clinical symptoms of relapse [27]. Despite the more common use of DIR and phase-sensitive inversion recovery MR sequences [28], active cortical lesions are still difficult to determine during the clinical routine, thus this association cannot be fully addressed in our study. In general, diffuse neuro-inflammation has moved into focus of MS pathophysiology and progressive diseases phases, but is still difficult to be fully revealed with conventional MRI techniques [29,30]. Positron emission tomography with high affinity ligands for microglia (e.g., TSPO) has been shown to reliably detect diffuse neuro-inflammatory processes [31], but is not feasible during the daily routine. A more practical approach provides quantitative MRI measurements which can detect CNS microstructural integrity [32].

The following limitations of our study have to be discussed. Although we implemented a standardized MRI protocol for the MS patients in our clinic, not all of our patients received these standardized MRI sequences due to unknown initial diagnosis during the presentation of symptoms or MRI capacity shortage. Moreover, Gd-enhancing lesions might be missed due to the limitations in the usage of Gd [33]. As mentioned before, a limited number of patients (18.6%) did not receive a cerebral and spinal MRI at the same time; however, detailed analyses of individual patients revealed that spinal cord lesions were quite unlikely in these patients. Furthermore, we conducted a retrospective design in a limited patient cohort which relied on the reevaluation of standardized clinical and radiological reports. Although diffusion-weighted images and 3D Double Inversion Recovery (DIR) sequences were included in our MRI protocol, lesions might still not have been recognized with our protocol, which might be further improved be the usage of myelin water fraction (MWF) and molecular proton fraction (MPF) mapping in future studies. In addition, volumetric measurements of lesions that also map slowly expanding lesions were not performed because we first aimed at evaluating our clinical questions by “conventional” methods, as performed during the daily clinical routine.

Our results have several implications for MS patients during the clinical practice. Our study shows that in spite of a clinical-radiological mismatch, most of the patients in group 2 were treated with MPS, though with a lower dose than MS patients with a clinical-radiological match (group 1). These patients profited similarly from the MPS treatment by an amelioration of their symptoms compared to the patients in group 1. This fact could again point towards an active inflammation/process that might not be completely unveiled with conventional MRI techniques. Besides the acute treatment of relapse, the definition of a new relapse also affects the current treatment choice and a possible escalation to a highly effective DMT [13]. In addition, the efficacy of current treatments has been categorized in clinical trials by mostly defining the primary endpoint as the annualized relapse rate, evaluated by clinical symptoms without the parallel assessment of MRI changes. With the more common application of NEDA criteria [34] during the clinical routine, MRI criteria have become more relevant for the decision of a breakthrough disease. However, NEDA criteria define “no evidence for diseases activity” and not disease activity itself. Thus, a standardized definition of relapse should be implemented including MRI and combinations of clinical symptoms with MRI activity and/or other biomarkers such as Neurofilament light.

In contrast to the clinical-radiological mismatch described in our study, the clinical-radiological paradox in MS patients is a well-recognized dissociation between MRI lesion load and clinical disability [15]. On MRI, there are either more lesions than expected from the clinical assessment or fewer lesions than anticipated by considerable clinical deficits. The clinical-radiological paradox could be at least partially explained by diffuse neuro-inflammatory processes as well. However, when compared to our study, this phenomenon does not take into account the temporal correlation between new symptoms and active MRI lesions, as studied in our patient collective.

## 5. Conclusions

The clinical-radiological mismatch during relapse might reflect different aspects during MS pathophysiology, possibly including diffuse neuro-inflammation mechanisms during progressive disease phases. Further clinical studies in larger patient collectives with highly evolved, standardized MRI protocols including volumetric lesion mapping are necessary to further address the mismatch between clinical relapse and active MRI lesion detection. From the clinical perspective, clinical relapses should be validated more consistently by defined MRI standards and additional biomarkers.

## Figures and Tables

**Figure 1 jcm-12-00739-f001:**
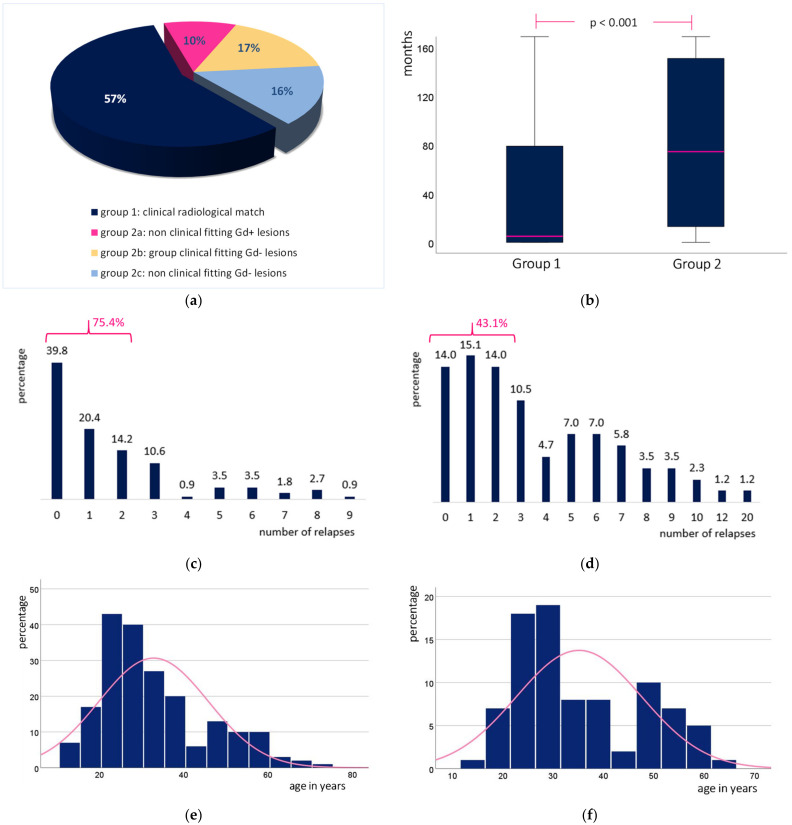
(**a**) Patient distribution. (**b**) Disease duration. (**c**) Number of previous relapses in group 1. (**d**) Number of previous relapses in group 2. (**e**) Age on acute relapse in group 1. (**f**) Age on acute relapse in group 2.

**Figure 2 jcm-12-00739-f002:**
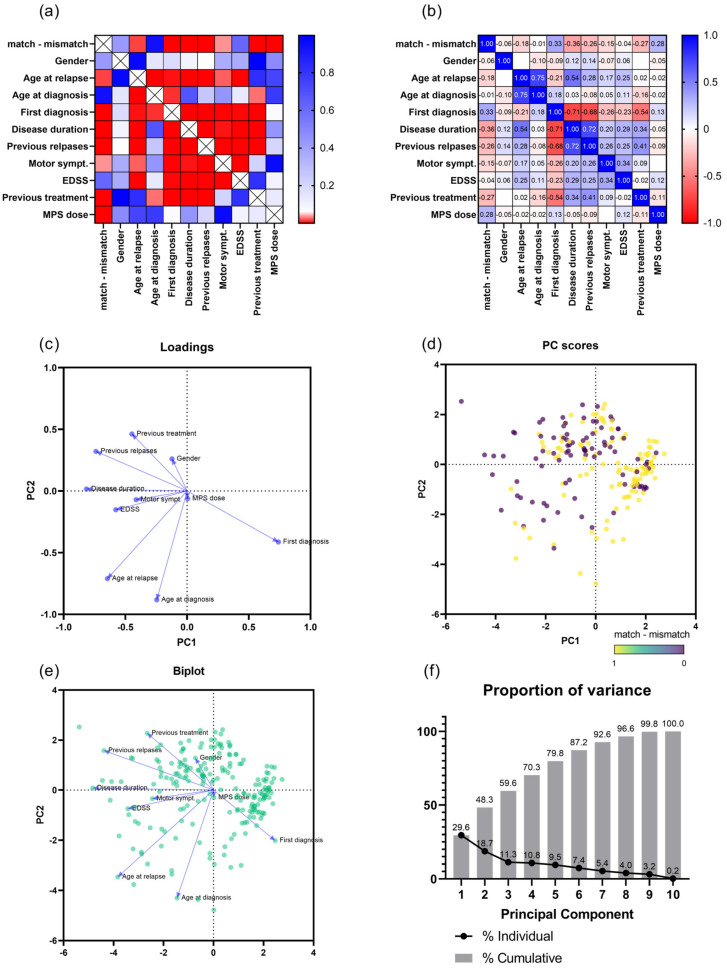
(**a**) Correlation matrix, *p* values are color coded, significant correlations are marked in red. (**b**) Correlation matrix, r values are displayed within squares and color coded according to positive correlations (blue) and negative correlations (red). (**c**) Principal component analysis, graphical representation of contributing factors to principal component 1 (PC1) and 2 (PC2). (**d**) Distribution of patients with clinical-radiological match (yellow dots) and clinical-radiological mismatch (purple dots) according to PC1 and PC2. (**e**) Merged representation of patients and the contributing factors for PC1 and PC2. (**f**) Proportion of variance, explained by principal components, PC1 and PC2 explain 48.3% of variance. Abbreviations: symp. = symptoms, MPS = methyl prednisolone, EDSS = Expanded Disability Status Scale, PC = principal component.

**Table 1 jcm-12-00739-t001:** Demographic and clinical data.

		All *n* = 199	Group 1 *n* = 113 (57%)	Group 2 *n* = 86 (43%)	*p* Value Mismatch vs. Non-Mismatch
**Sex (n, female)**		145 (73%)	80 (71%)	65 (76%)	n.s.
**Age at diagnosis (years)**		28.0	27.9	28.1	n.s.
**Age at current relapse (years)**		32.9	31.2	35.2	0.013
**Diagnosis of MS with current relapse**		41%	55%	22%	<0.001
**No of relapses before </= 2**		60.8%	75.4%	43.1%	<0.001
**Disease duration (months)**		71.3	45.6	105.1	<0.001
**EDSS in current relapse < 3.0**		57.8%	58.3%	57.0%	n.s.
**Functional system concerned**					
	brainstem/ cranial nerves	39.2%	43.4%	33.7%	n.s.
	motoric dysfunction	31.7%	25.7%	39.5%	0.038
	sensory	54.8%	51.3%	59.3%	n.s.
	coordination/ataxia	12.6%	12.4%	12.8%	n.s.
	bowel/bladder	4.0%	3.5%	4.7%	n.s.
	others	1.5%	1.8%	1.2%	n.s.
**Treatment with MPS**		95%	98%	91%	n.s.
	3 g	34.7%	24.8%	47.7%	<0.001
	5 g	40.2%	52.2%	24.4%	<0.001
	15 g	6.5%	8.0%	4.7%	n.s.
	other doses	18.6%	15.0%	23.2%	n.s.
**DMT**		52.5%	45.1%	58.2%	n.s.
**MRI with 3.0 T**		43.2%	43.4%	43.0%	n.s.

Abbreviations: n.s. = not significant, MS = multiple sclerosis, EDSS = Expanded Disability Status Scale, MPS = methyl prednisolone, DMT = disease modifying therapy, MRI = magnetic resonance imaging.

## Data Availability

The data presented in this study are available on request from the corresponding author. The data are not publicly available due to ethical regulations in our clinic.

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
