# Peer review of "Clinical-Radiological Mismatch in Multiple Sclerosis Patients during Acute Relapse: Discrepancy between Clinical Symptoms and Active, Topographically Fitting MRI Lesions"

_jcm, 2023, doi:10.3390/jcm12030739_

Round 1

Reviewer 1 Report

This retrospective study presents an analysis of the correspondence between the clinical symptoms of MS relapse and the topography of active CNS lesion. The manuscript describes the current issues of clinical application and validation of MRI protocols for instrumental assessment of active lesions in clinical relapses of MS.

The topic described in this manuscript is actual and interesting. The manuscript is well organized and written.

 Please see the minor comments below:

 1. The authors state in lines 264-266 “around 20% of patients did not receive standardized MRI sequences”. How were these patients distributed between subgroups 1, 2 (a, b, c)? 

 2. Figure 2. (a) Patient distribution - please also indicate the subgroups of patients (group 1, 2a, 2b, 2c) in the diagram caption. This will make the diagram and text on lines 178-189 easier to read.

 3. The Discussion section provides a detailed discussion of the findings and limitations of the study. However, the authors do not mention the limitations of conventional MRI techniques which are mostly can detect only clearly visible demyelinating lesions in contrast to methods of myelin quantification - diffusion tensor imaging or non-conventional methods, such as myelin water fraction (MWF), molecular proton fraction (MPF) mapping.

Author Response

Reviewer 1

This retrospective study presents an analysis of the correspondence between the clinical symptoms of MS relapse and the topography of active CNS lesion. The manuscript describes the current issues of clinical application and validation of MRI protocols for instrumental assessment of active lesions in clinical relapses of MS.

The topic described in this manuscript is actual and interesting. The manuscript is well organized and written.

 Please see the minor comments below:

  1. The authors state in lines 264-266 “around 20% of patients did not receive standardized MRI sequences”. How were these patients distributed between subgroups 1, 2 (a, b, c)? 

We thank the reviewer for this hint we observed the following distribution (page 4):

Although we implemented a standardized MRI protocol for the MS patients in our clinic, 24 of our patients (12.1%) did not receive these standardized MRI sequences mostly due to unknown initial diagnosis. In detail, 14.2% of these patients were found in group 1, 2 patients (2.3%) in group 2a, 5 patients (5.8%) in group 2b and 1 patient (1.2%) in group.

  1. Figure 2. (a) Patient distribution - please also indicate the subgroups of patients (group 1, 2a, 2b, 2c) in the diagram caption. This will make the diagram and text on lines 178-189 easier to read.

We thank the reviewer for this suggestion and changed the diagram caption accordingly.

  1. The Discussion section provides a detailed discussion of the findings and limitations of the study. However, the authors do not mention the limitations of conventional MRI techniques which are mostly can detect only clearly visible demyelinating lesions in contrast to methods of myelin quantification - diffusion tensor imaging or non-conventional methods, such as myelin water fraction (MWF), molecular proton fraction (MPF) mapping.

We agree with the reviewer that MRI limitations should be pointed out more clearly. We therefore added the following sentence in the discussion section (page 12):

Although diffusion weighted images and 3D Double Inversion Recovery (DIR) sequences were included in our MRI protocol, lesions might still not have been recognized with our protocol which might be further improved be the usage of myelin water fraction (MWF) and molecular proton fraction (MPF) mapping in future studies.

Reviewer 2 Report

The manuscript title: “Clinical-radiological mismatch in multiple sclerosis patients 2 during acute relapse: discrepancy between clinical symptoms 3 and active, topographically fitting MRI lesions”

Referee’s Comments:

Multiple sclerosis is the most common immune-mediated disorder affecting the central nervous system. More than 3.5 million people are affected globally per year by this disease. Treatments attempt to improve function after an attack and prevent new attacks. Unfortunately, there is no known cure for multiple sclerosis disease. All mentioned above confirms the importance of the investigations performed.

The manuscript is organized in a clear manner. The number of the patients used in the study is sufficient, however this does not refer to the cited authors, and also to the methods applied. Only one table and one figure in the whole manuscript are not enough. Figure 1 is mentioned on Lines 136 and 142, however I cannot see where is this figure in the manuscript, I see only Figure 2 on Line 192. I do recommend more literature data on diffuse neuroinflammation and their analysis to be included. Also more parameters should be studied, for example by means of principal component analysis, and/or hierarchical cluster analysis in order to obtain more profound results and conclusions.

Author Response

Reviewer 2

Multiple sclerosis is the most common immune-mediated disorder affecting the central nervous system. More than 3.5 million people are affected globally per year by this disease. Treatments attempt to improve function after an attack and prevent new attacks. Unfortunately, there is no known cure for multiple sclerosis disease. All mentioned above confirms the importance of the investigations performed.

The manuscript is organized in a clear manner. The number of the patients used in the study is sufficient, however this does not refer to the cited authors, and also to the methods applied. Only one table and one figure in the whole manuscript are not enough. Figure 1 is mentioned on Lines 136 and 142, however I cannot see where is this figure in the manuscript, I see only Figure 2 on Line 192. I do recommend more literature data on diffuse neuroinflammation and their analysis to be included. Also more parameters should be studied, for example by means of principal component analysis, and/or hierarchical cluster analysis in order to obtain more profound results and conclusions.

In summary, I recommend publication of the manuscript after major revision.

We thank the reviewer for has thoughtful comments. We first apologize for mislabeling of Figure 1 and corrected accordingly. We also included more literature on diffuse neuro-inflammation and added the following paragraph (page 12):

In general, diffuse neuro-inflammation has moved into focus of MS pathophysiology and progressive diseases phases but is still difficult to be fully revealed with conventional MRI techniques [29,30]. Positron emission tomography with high affinity ligands for microglia (e.g. TSPO) has been shown to reliably detect diffuse neuro-inflammatory processes [31] but is not feasible during the daily routine. A more practical approach provide quantitative MRI measurements which allow to detect CNS microstructural integrity [32].

As stated in additional comments (and also manuscript page 10) our primary aim was to evaluate our clinical observation of a significant clinical-radiological mismatch during relapse using “conventional” MRI techniques. For future studies, more advanced volumetric and quantitative MRI techniques should be applied to further dissect diffuse neuro-inflammation which is now stated in the discussion section (also see other reviewer)

We also performed additional multivariate analyses in order to further dissect major contributors for the clinical radiological mismatch. We now include a whole new Figure 2 with correlation matrix and principle component analysis and also added the following parts:

Page 4:

Correlation matrix (Spearman) and principal component analysis was performed to further determine the most important factors that would discriminate between group 1 and group 2 (dependent variable) including the following clinical factors: gender, age at relapse, age at diagnosis, first diagnosis (during current relapse), disease duration, previous relapse, previous treatment, motor symptoms (only significant symptom), EDSS, previous treatment and MPS dosage. Two principal components (PCs) were extracted based on parallel analysis.    

Page 9:

 In order to further weight and group the influence of the clinical factors to the differentiation into group 1 and group2, we additionally performed correlation and principal component analyses (Figure 2, supplementary tables). A significant correlation with group 1 / group 2 (match – mismatch) and age at relapse, first diagnosis (at current relapse), dis-ease duration, previous relapse, motoric symptoms, previous treatment and MPS dosage was detectable, however, r values were all below 0.4. Multiple other correlations were found between the other clinical factors such as disease duration and previous relapse, further details are displayed in Figure 2a and 2b. Two principal components (PC1 and PC2) were extracted in our PC model that explain 48.3% of variance (PC1 – 29.6%, PCA2 – 18.7%). Major contributors for PC1 were disease duration, first diagnosis (current relapse lead the diagnosis of MS) and previous relapse; for PC2 major factors were age at diagnosis, age at relapse and previous treatment.           

Page 12:

The results from our principal component analysis further support this observation since age at relapse, age at diagnosis and first diagnosis at current relapse were major factors in the differentiation between group 1 and 2.    

Reviewer 3 Report

Journal Article: Clinical-radiological mismatch in multiple sclerosis patients during acute relapse: discrepancy between clinical symptoms and active, topographically fitting MRI lesions

Dunschede et al.

Overview: This is an observational, retrospective analysis of 199 MS patients who developed neurological symptoms that were interpreted as consistent with a clinical relapse who also underwent clinical MRI. Patients were classified based on clear radiological evidence of an acute MS lesion that was in a region that would explain the clinical syndrome of the patient compared to patients without corresponding lesions. Subgroup analysis of the latter group looked at those with new lesions in non-anatomically corresponding areas, old lesions in anatomically corresponding regions, and patients with neither of the above. The main observation was that 57% of clinical relapses had corresponding acute MS lesions. 

The overall study design and focus on clinical characteristics of their cohorts provides some interesting observations particularly in the cohorts without corresponding acute lesions. However, as these subgroup analyses were from a much smaller groups and is a major limitation for of the study. While I find their study interesting, I think it also lacks interest to a general medical/science audience and is more suitable for a MS or Neurology Journal. 

Minor Comments:

Methods: 

1.     Providing data on the delay in time from clinical symptoms to clinical MRI that was analyzed may provide useful information as long or short delays in time could affect the ability to identify new enhancing lesions

2.     What is the resolution that they acquired their MRIs at?

3.     “Standardized MRI reports were reanalyzed by correlating clinical 92 symptoms and their anatomical representation in the human brain .. “ It is not clear if the images were reanalyzed or just the reports. If the images were reexamined, by whom and what experience/expertise do they have.

Results:

1.     “A marked patient reported improvement of clinical symptoms following MPS 162 treatment was evident in both groups with 90% in group 1 and 84% in group 2.” – It is not clear how this result was determined. 

2.     The comparison between fitting and non-fitting cohorts, was lesion size taken into consideration? 

3.     It may be helpful to provide information on the baseline MRI characteristics of their cohorts (lesion volume/number) as well as characteristics of the CELs (number/volume).

Author Response

Reviewer 3

Overview: This is an observational, retrospective analysis of 199 MS patients who developed neurological symptoms that were interpreted as consistent with a clinical relapse who also underwent clinical MRI. Patients were classified based on clear radiological evidence of an acute MS lesion that was in a region that would explain the clinical syndrome of the patient compared to patients without corresponding lesions. Subgroup analysis of the latter group looked at those with new lesions in non-anatomically corresponding areas, old lesions in anatomically corresponding regions, and patients with neither of the above. The main observation was that 57% of clinical relapses had corresponding acute MS lesions. 

The overall study design and focus on clinical characteristics of their cohorts provides some interesting observations particularly in the cohorts without corresponding acute lesions. However, as these subgroup analyses were from a much smaller groups and is a major limitation for of the study. While I find their study interesting, I think it also lacks interest to a general medical/science audience and is more suitable for a MS or Neurology Journal. 

Minor Comments:

Methods: 

  1. Providing data on the delay in time from clinical symptoms to clinical MRI that was analyzed may provide useful information as long or short delays in time could affect the ability to identify new enhancing lesions

We thank the reviewer for this important comment and provide the following additional information (page 3):

MRIs were taken within 17 days in average after the first symptoms occurred, were acquired with 3 T and 1.5 T scanners…

The time between first symptoms and MRI acquisition was 18 days in group 1 and 17 day in group 2 (no significant difference).

  1. What is the resolution that they acquired their MRIs at?

We thank the reviewer for this hint and added the following information (page 3):

MRIs were taken within 17 days in average after the first symptoms occurred and acquired with 3 T and 1.5 T scanners according to a standardized protocol that consists of a native 3D T1 MPRAGE (1 mm isotropic, 3T: TR/TI 2300/900ms, FA 8°, 1.5T: TR/TI 1280/660ms FA 15°), 3D Double Inversion Recovery (DIR, TR/TI1/TI2 7500/3000/450 ms, 1.5T: 1.33 mm isotropic, TE 337ms, 3T: 1mm isotropic, TE 392ms), post contrast 3D T2-FLAIR (1 mm isotropic, 3T: TR/TI/TE 7000/2050/392ms, 1.5T: TR/TI/TE 5000/1800/337ms) and T1 MPRAGE (same protocol as precontrast), as well as additional diffusion weighted images and 3mm axial T2-TSE sequences that cover the posterior fossa and orbit (in-plane resolution 3T: 0.6 x 0.5 mm², 1.5T: 0.9 x 0.7mm²).

  1. “Standardized MRI reports were reanalyzed by correlating clinical 92 symptoms and their anatomical representation in the human brain .. “ It is not clear if the images were reanalyzed or just the reports. If the images were reexamined, by whom and what experience/expertise do they have.

We agree with the reviewer that this important point should be described more clearly, we therefore added the following sentences (page 3):

These MRI reports were reanalyzed by correlating clinical symptoms and their anatomical representation in the human brain [9,10] with new gadolinium enhancing (active) lesions by an experienced neurologist. Whenever this description was unclear or a discrepancy between clinical symptoms and MRI lesions was found, original MRI images were reanalyzed by two experienced neuro-radiologists to validate the resultsThese MRI reports were reanalyzed by correlating clinical symptoms and their anatomical representation in the human brain [9,10] with new gadolinium enhancing (active) lesions by an experienced neurologist. Whenever this description was unclear or a discrepancy between clinical symptoms and MRI lesions was found, original MRI images were reanalyzed by two experienced neuro-radiologists to validate the results.

Results:

  1. “A marked patient reported improvement of clinical symptoms following MPS 162 treatment was evident in both groups with 90% in group 1 and 84% in group 2.” – It is not clear how this result was determined. 

We thank the reviewer for this hint, the improvement was evaluated according to the patient reported outcome and clinical examination after application of MPS. We would like to clarify as follows (page 6):

A marked improvement of clinical symptoms following MPS treatment, defined as patient reported outcome and / or improved clinical examination according to the final reports, was evident in both groups with 90% in group 1 and 84% in group 2.

  1. The comparison between fitting and non-fitting cohorts, was lesion size taken into consideration? 

We thank the reviewer for this important comment. Lesion size was not taken into account but would be very interesting in the context of standardized volumetric assessments of MS brain lesions. The primary aim of our study was to first evaluate our observation that clinical symptoms during relapse do often not associated with active MR lesions in MS in the daily routine. In order to test this hypothesis, we applied “conventional” MRI evaluation as performed during the clinical routine. However, we totally agree that more advanced imaging tools in a bigger and prospective study would be highly interesting to further study our question. To further address this point in the manuscript, we added the following sentence in the discussion section (also addressing comment below):

Page 13:

In addition, volumetric measurements of lesions also mapping slowly expanding lesions were not performed since we first aimed at evaluating our clinical questions by “conventional” methods as performed during the daily clinical routine. 

Page 13:

Further clinical studies in larger patient collectives with highly evolved, standardized MRI protocols including volumetric lesion mapping are necessary to further address the mismatch between clinical relapse and active MRI lesion detection.

  1. It may be helpful to provide information on the baseline MRI characteristics of their cohorts (lesion volume/number) as well as characteristics of the CELs (number/volume).

We agree with the reviewer that MRI baseline characteristics could be interesting. However, 41% of the patients did not receive a previous MRI scan and MS diagnosis was established with the current clinical assessment. Since there was a huge discrepancy between patients with MS diagnosis at the current relapse (group 1 - 55% versus group 2 – 22%, see table 1) we did not assess MRI baseline characteristics for further comparisons. We also agree that volumetric lesion measurements would be highly interesting and possibly more valid for baseline MRI assessments than lesion counts, however, this was beyond the scope of this study. We addressed this comment in the manuscript together with the point above.